# Performance of prototype serological immunoassays for foot-and-mouth disease virus using G-H loop peptides and stabilized virus-like particles

Abdelaziz A. Yassin,[1,2,3] Yvonne Sewell,[1] Anna B. Ludi,[1] Alison Burman,[1] Georgina Limon,[1] Madeeha Afzal,[1] Daniel Horton,[2,4] Donald P. King,[1] Amin S. Asfor[1,2,4]

**ABSTRACT**    Inter-serotype cross-reactivity of foot-and-mouth disease (FMD) antibody enzyme-linked immunosorbent assays (ELISAs) can exceed 50%, leading to incorrect serotyping of outbreaks with implications for vaccine selection. In this study, synthetic peptides that mimic the hypervariable G-H loop of FMD viruses (FMDVs) that currently circulate in East Africa (O, A, SAT1, and SAT2) were evaluated as capture antigens in ELISAs (pELISAs). A panel of monovalent bovine sera was tested using these novel assays in parallel with separate ELISAs that utilized stabilized virus-like particles (VLPs). Virus neutralization tests using the same viruses were used to benchmark the status of the sera, which revealed evidence of cross-reactivity for the serotype O and SAT2 antigens (encompassing 2/19 and 3/19 of the heterologous sera, respectively). Equivalent diagnostic serotype sensitivity was observed for prototype peptide and VLP ELISAs for serotype O and SAT1 antigens (86% and 100%, respectively), while there was higher diagnostic serotype sensitivity for the VLP ELISAs targeting serotypes A and SAT2 compared to the corresponding pELISAs (86% vs 71% and 100% vs 86%, respectively). The serotype specificity of these tests ranged from 71% to 79% and 52% to 89% for the pELISA and VLP ELISA formats, respectively. Peptides offer a simple, biosafe, and cost-effective approach to present FMDV-specific epitopes, and these initial findings suggest that peptide ELISAs could be a promising approach to develop serological ELISA assays to present authentic epitopes in comparison to ELISAs that use full capsid VLPs.

**IMPORTANCE** This study marks a significant advancement in the development of safe, affordable, and scalable diagnostic tools for foot-and-mouth disease virus. By employing the G-H loop, a critical epitope on the receptor-binding domain within the viral capsid, we demonstrate that the G-H loop peptide-based ELISAs can effectively mimic and present authentic epitopes as the full virus particles in serological assays. This approach offers several important benefits: **enhanced biosafety** is achieved by eliminating the need for whole virus particles, thereby reducing biosafety risks. **Cost-effectiveness**: through simplifying antigen production, enabling broader deployment in low and middle income (LMI) regions. **Serotype-specific accuracy**: tailored antibody ELISAs improve diagnostic accuracy and post-vaccination monitoring. It also improves the sensitivity and specificity when compared to the commercially available kits, which suffer from cross-reactivities (>50% in some cases). The performance of the developed ELISA is comparable to gold-standard virus neutralization tests as a benchmark, which underscores the reliability of the approach.

**KEYWORDS**    foot-and-mouth disease, ELISA, peptides, virus-like particle, SNT, G-H loop

Address correspondence to Amin S. Asfor, a.asfor@surrey.ac.uk.

The authors declare no conflict of interest.

See the funding table on p. 11.

Foot-and-mouth disease virus (FMDV) is a member of *Aphthovirus* genus in the order Picornavirales within the family Picornaviridae. There are seven FMDV serotypes: O, A, Asia 1, SAT1, SAT2, SAT3, and C, although serotype C has not been detected since 2004 (1). The most widespread serotype worldwide is O, followed by A. In contrast, the SAT serotypes are mainly found in Africa, while Asia 1 is distributed in Asia. There is no cross-protection between serotypes or even sometimes between lineages within a serotype.

Serological assays are required to support surveillance activities, certify the export and import of livestock, and in the evaluation of post-vaccinal responses (2). Diagnostic serological assays can be characterized into two groups: (i) non-structural protein (NSP) antibody tests, which are pan-serotypic and can be used to identify infected animals whether or not vaccination has been used (so long as the vaccine has been purified to remove NSPs) and (ii) structural protein (SP) ELISAs and virus neutralization tests (VNTs), which are serotype-specific tests and cannot differentiate between infected and vaccinated animals but instead are used to assess the antibody response after infection and vaccination.

The VNT measures neutralizing serotype-specific antibodies and titers have been correlated to protection (3, 4). However, the VNT is difficult to carry out due to its dependence on cell culture, passaging of the virus, the technicality of performing the test, and the need for a high containment laboratory. Also, as it is dependent on cells, VNT results can be highly variable. In contrast, ELISAs can be performed in a few hours, do not require high containment laboratories, and are technically not as challenging (3, 5, 6). ELISAs are also considered to be highly reproducible and have been correlated to protection for the South American FMDV strains (7, 8). The difficulty with using SP-ELISAs is that they exhibit inter-serotype cross-reactivity (i.e., generate false positive results for serotypes that the animal has not been exposed to). This cross-reactivity can constrain the use of these ELISAs for surveillance in endemic countries or post-vaccination monitoring, where multiple serotypes are present (9–12). Cross-reactivity has been documented when monovalent sera were tested by in-house and commercial ELISA kits, where false positive results can exceed 50% between serotypes (13). It is assumed that these inter-serotype cross-reactive signals are due to the presence of common epitopes that are shared across the different serotypes.

The FMDV capsid is fragile and can readily dissociate into component pentamers after heating or treatment at low pH. Density sedimentation is used to identify intact virions that have a sedimentation coefficient (S) of 146S compared to dissociated pentamers that sediment at 12S (14, 15). Virus-like particles (VLPs) lack genomic RNA and sediment at 75S (16, 17). The structural proteins (VP1, VP2, VP3, and VP4) that form the capsid contain serotype-specific and cross-reactive epitopes, which can either be neutralizing or non-neutralizing (18). Five antigenic sites have been identified for serotype O, including site 1, which encompasses the G-H loop and the C-terminus of VP1 (19, 20). Site 1 has also been described for serotypes A (21, 22) SAT1, and SAT2 (23). The length of VP1 is very variable between serotypes due to insertions or deletions mainly in the region around the G-H loop (24). The G-H loop is a highly flexible, surface-exposed region of the capsid, and its conformation differs slightly between serotypes, influencing the range of loop lengths that can be structurally accommodated (25, 26). Comparative genomic analyses show that SAT serotypes, in particular, exhibit length variability in this region, largely due to their distinct evolutionary histories and circulation in wildlife reservoirs (24, 27). The G-H loop includes the receptor-binding motif (arginine-glycine-aspartic acid [RGD]), which binds to the integrin receptor on host cells (28, 29). More than 25% of neutralizing antibodies are thought to be directed toward this loop (30). The G-H loop contains overlapping continuous linear epitopes (31–33).

This manuscript focuses on the evaluation of novel serological ELISAs tailored for East Africa, a region where FMDV is endemic as this is an area endemic to FMDV. Outbreaks in this region are due to four serotypes (O, A, SAT1, and SAT2). We therefore compared the reactivity of monovalent sera with known provenance, collected from infected and

vaccinated animals, for reactivity in G-H loop-based peptide ELISAs (pELISAs) and a corresponding VLP-based ELISA. The serotype sensitivity (the proportion of homologous sera that were correctly detected using the peptides or VLPs) and the serotype specificity (the proportion of heterologous sera that did not react against the peptides or VLPs) were compared with the results from the virus neutralization test.

## MATERIALS AND METHODS

### Cell, virus propagation, and virus neutralization test

VNTs were carried out to test the ability of monovalent sera for the four serotypes to neutralize the East African FMDV isolates. The IBRS-2 (pig kidney) cell line (34) was used for FMDV propagation and VNT. Cells were maintained either in Dulbecco's modified Eagle's medium or in Dulbecco's minimum essential medium (Thermo-Fisher Scientific, United Kingdom) supplemented with 10% heat-inactivated adult bovine serum (Thermo-Fisher Scientific, United Kingdom).

Four viral isolates (O KEN/4/2018, A SUD/9/2018, SAT1 TAN/22/2014, and SAT2 KEN/19/2017) were neutralized against the monovalent sera with an initial dilution of 1/16 in twofold serial dilution. The results were reported as the final dilution required to neutralize 50% of the inoculated cultures (3) between 32 and 320 (1.5 $\log_{10}$ to 2.5 $\log_{10}$) 50% tissue culture infectious dose ($TCID_{50}$) virus doses. As stated by the World Organisation for Animal Health, antibody titers below 1/45 (i.e., 1.65 $\log_{10}$) of the final serum dilution were regarded as negative.

### Peptide synthesis

Four peptides (31–35 mers) were designed based on the full-length VP1 G-H loops corresponding to FMDVs in East Africa based on the following isolates: O KEN/4/2018, A SUD/9/2018, SAT1 TAN/22/2014, and SAT2 KEN/19/2017 (accession numbers MT602084, MT602079, MT602088, and MT602092) (Table 1). The differences in the peptide lengths used reflect the differences in the conformation of the G-H loop between different serotypes. Six lysine residues were added at the C-terminus to increase the solubility of these peptides. All peptides were synthesized at Peptide Protein Research Ltd, United Kingdom.

### Production of virus-like particles

VLPs were produced using a vaccinia virus expression system as previously described (35) for the following FMDV isolates: O KEN/4/2018, A SUD/9/2018, SAT1 TAN/22/2014, and SAT2 KEN/19/2017 sequences (accession numbers MT602084, MT602079, MT602088, and MT602092). Sequences were obtained from GenBank and modified by substituting nucleotides encoding a cysteine residue (TGC codon) at amino acid site 93 in VP1 to improve capsid stability (35, 36), and the corresponding peptides were synthesized at GeneArt, Thermo Scientific, United Kingdom. The recombinant VLPs were purified through a sucrose cushion and sucrose density centrifugation. The peak fractions containing purified VLPs were identified by SDS-PAGE, then aliquoted and stored at 4°C. Zeba Spin Desalting Columns (Thermo Scientific, United Kingdom) were then used to exchange the buffers from sucrose to phosphate buffer saline. For electron microscopy,

**TABLE 1** East African peptides used in this study[a]

| Parent virus | Amino acid sequence |
| --- | --- |
| O KEN/4/2018 | CRYSSAPATNV**RGD**LQVLAQRVARTKKKKKK |
| A SUD/9/2018 | TTKYTADTPPR**RGD**LGALAARLAAQKKKKKK |
| SAT1 TAN/22/2014 | YKPTSEAPRTNI**RGD**LATLAERIASEKKKKKK |
| SAT2 KEN/19/2017 | NGECKYTDRVSAI**RGD**RTVLAAKYADSRHKKKKKK |

[a]Six lysine residues were added at the C-end of each peptide to increase solubility. RGD is in boldface to highligt the central position of RGD during the peptides design.

VLPs were concentrated using Amicon Ultra filtration devices 100K (MilliporeSigma, United Kingdom). VLPs were stored at 4°C until tested.

## Negative-stain transmission electron microscopy (TEM)

Electron microscopy was used to confirm the presence of assembled VLP particles with expected intact FMDV capsids. Seven microliters of the purified and concentrated sample was placed on glow-discharged, formvar/carbon-coated copper TEM grids (Agar Scientific, Stansted, United Kingdom) and negatively stained for 1 minute using 2% aqueous uranyl acetate. The prepared grids were imaged at 100 kV in an FEI Tecnai 12 TEM with Tietz F214 CCD camera.

## Serum samples

Thirty-two bovine sera, seven for serotypes A, O, and SAT2 and five for SAT1, were selected from animals infected and/or vaccinated with a single FMDV serotype (i.e., monovalent) representing O, A, SAT1, or SAT2 (Table S1). Prior to use, sera were heat inactivated at 56°C for 30 minutes and stored at −20°C. Six bovine serum samples sourced from a country officially free from FMD disease were included as negative control samples.

## Peptide and VLP ELISAs

Indirect pELISAs were developed using the peptides listed in Table 1. To determine the optimal concentration of the peptides, different concentrations, ranging from 10 to 1.25 µg/mL, were initially evaluated. The optimum coating concentration for the pELISAs was found to be 5 µg/mL (data not shown). For the VLP ELISA, different concentrations of the VLPs were tested, ranging from 4 to 0.125 µg/mL, where the optimum concentration was found to be 2 µg/mL (data not shown).

Briefly, plastic 96-well plates (Maxisorp; Nunc) were coated with 50 µL per well of the peptides in 0.05 M carbonate/bicarbonate coating buffer (pH 9.6) at 4°C overnight. For the VLP ELISA, a recombinant bovine alphaVbeta6 integrin produced at The Pirbright Institute was used at the concentration of 1 µg/mL to coat the plates (37–39). Wells were then washed three times with phosphate-buffered saline (PBS) containing 0.1% Tween 20 and patted dry. This washing step was repeated between all incubation steps. Wells were then blocked with 200 µL blocking buffer (5% [wt/vol] skimmed milk-PBS-1% horse serum (NZ 16050122, Life Technologies) at 37°C for 1 hour. For the integrin-coated plates, VLPs were added and incubated at 37°C for 1 hour. The plates were then incubated (37°C for 1 hour) with 50 µL of the test sera pre-diluted to 1/25 in duplicates. Then 50 µL of species-specific horseradish peroxidase (HRP)-conjugated secondary antibodies was diluted to 1/15,000 for anti-bovine-IgG conjugate (A18751) (Life Technologies Ltd, United Kingdom) in dilution buffer in 1% (wt/vol) skimmed milk-PBS. The chromogen development was mediated by the addition of 50 µL of HRP substrate (3,3′,5,5′-tetramethylbenzidine) (TMBW-0100-01, Sigma FAST; Sigma, United Kingdom). The reaction was stopped after 15 minutes by the addition of 50 µL of 1 M sulfuric acid, and the optical density (OD) was measured at 450 nm using a SpectraMax ABS plate reader (Molecular Devices, LLC, United States).

## Statistical analysis for the assays (cutoff, serotype sensitivity, and serotype specificity)

Receiver operating characteristic (ROC) analysis was carried out using the Wilson and Brown method (40, 41) to determine the optimal cutoff for the best serotype sensitivity and serotype specificity in each of the pELISAs and VLP ELISAs using GraphPad Prism (Version 9. 4. 1 [681]).

Serotype sensitivity was calculated as the proportion of homologous sera (one serotype for all the settings) that reacted against the peptides, VLPs, or VNT. Serotype

specificity was calculated as the proportion of heterologous (serotype other than the tested serotype) sera that did not react against the peptides, VLPs, or in the VNT.

## RESULTS

### Neutralization responses of monovalent bovine sera

The bovine sera had a known provenance with respect to FMDV serotype that the cattle had been exposed to, and this information was used to define the expected outcomes from the testing by VNT and ELISA. For the VNT, the highest diagnostic serotype sensitivity was seen for O KEN/4/2018 with 100% of the serotype O sera generating positive titers higher than 1.65 $\log_{10}$. These same sera were negative for the other FMDV antigens apart from two weak positive responses (1.65 $\log_{10}$) that were generated using the SAT2 KEN/19/2017 isolate (Fig. 1). For the sera representing the other FMDV serotypes, it was not always possible to detect homologous responses, and there was more inter-serotype cross-reactivity evident. For example, only 4/7 of the serotype A sera generated titers above the test cutoff, and one of these sera gave a titer of 1.80 $\log_{10}$ using the SAT2 KEN/19/2017 isolate. Testing of the monovalent SAT1 sera generated homologous titers for 4/5 of the samples, while two samples were positive using the serotype O antigen. No cross-serotype reactivity was observed for the serotype SAT2 sera, where 6/7 were positive for the homologous antigen.

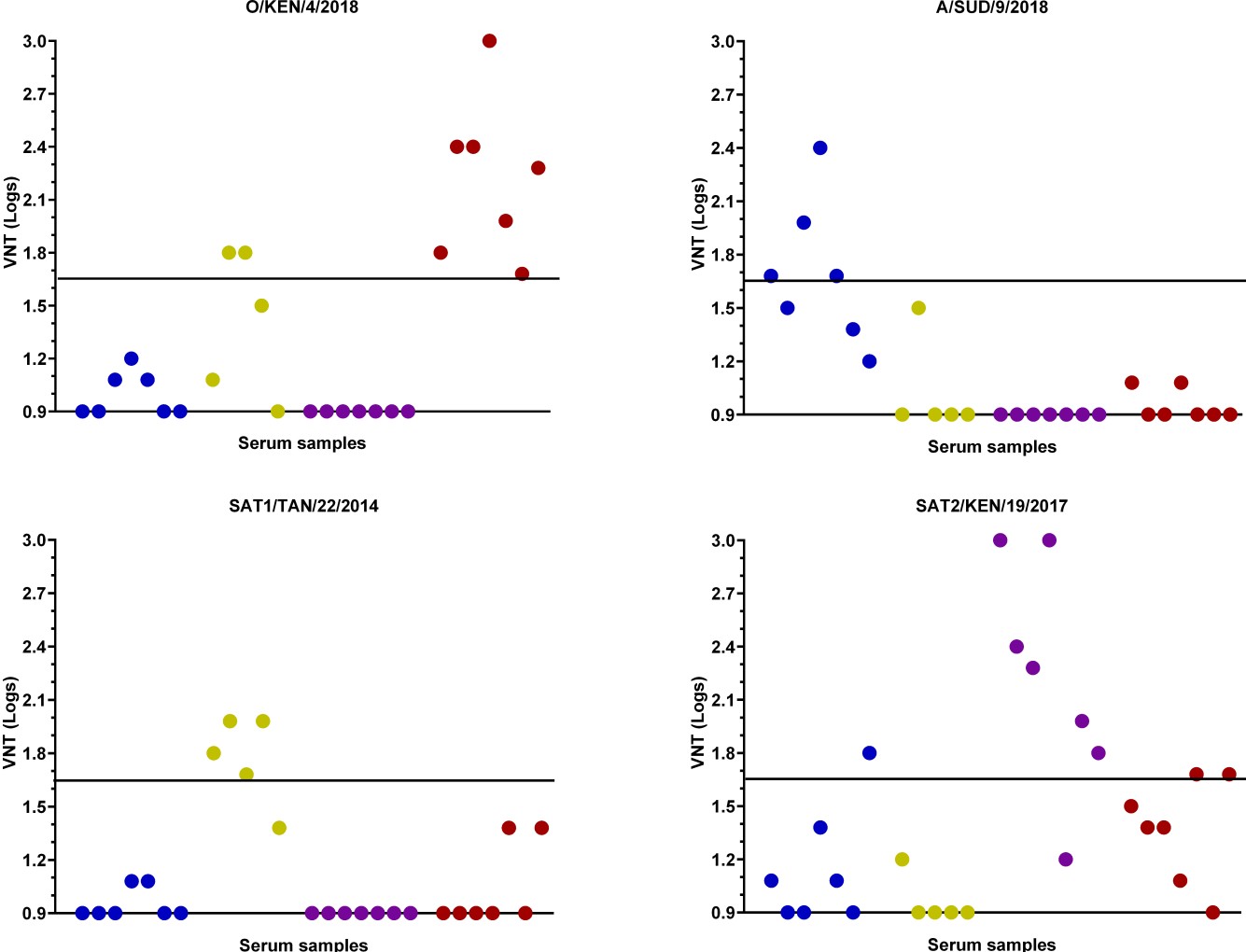

**FIG 1**  VNT testing monovalent sera against East African FMDV isolates O KEN/4/2018, A SUD/9/2018, SAT1 TAN/22/2014, and SAT2 KEN/19/2017. The colors highlight the serotype provenance of the monovalent sera: O (red), A (blue), SAT1 (yellow), and SAT2 (purple). The black line denotes the test cutoff (1.65 $\log_{10}$).

## Assessment of VLPs integrity by electron microscopy

Peak fractions from sucrose density centrifugation were assessed using negative stain electron microscopy. Fraction 11 for O KEN/4/2018, A SUD/9/2018, and SAT2 KEN/19/2017 and fraction 12 for SAT1 TAN/22/2014 were examined, showing the presence of 75S and 12S FMDV particle components (Fig. 2).

## Reactivity of sera against peptides and VLPs

A set of 32 monovalent cattle sera was tested in ELISA using the peptides and VLPs as antigens (Table 2). According to the ROC analyses, the optimum serum dilution was 1/25 for the pELISAs, while 1/125 was optimum for the VLP ELISAs (Fig. S1). After calculating the cutoff using the ROC curves, the serotype sensitivity and serotype specificity were determined for each serotype in each assay (Fig. 3 and 4). Using these parameters, the highest diagnostic serotype sensitivity for the pELISAs was 100% and was seen for the SAT1 TAN/22/2014 antigen, with cross-reactivity observed for five O sera and a single A serum. The corresponding VLP ELISA also had 100% serotype sensitivity, while there was cross-reactivity for 10 sera (four for serotype A and three each for serotypes O and SAT2, respectively). The serotype sensitivity for the serotype O pELISA and VLP ELISA was both 86%. Four serotype SAT1 sera cross-reacted with serotype O pELISA, and three serotype A sera cross-reacted with VLP ELISA. Six of seven homologous serotype SAT2 sera reacted in the SAT2 pELISA, while the serotype sensitivity for the corresponding VLP ELISA was 100%. Cross-reactivity for the SAT2 pELISA was seen for four sera (one SAT1 and 3 O), while four sera also cross-reacted with the SAT2 VLP ELISA (one each for A and SAT1 and two SAT2). The serotype A antigen tests exhibited a serotype sensitivity of 71% for the pELISA and 86% for VLP ELISA. There was heterologous cross-reactivity for four sera in the pELISA (three for SAT1 and one for O) and two sera in the VLP test (one each for SAT1 and O). The serotype specificity of these tests was defined by the proportion of heterologous sera that did not react in the VLP and peptide ELISAs. The lowest observed serotype specificity of 71% for the pELISAs was for SAT1, where 6/21 heterologous sera cross-reacted to the peptide (1/7 for A and 5/7 for O). The corresponding SAT1 VLP ELISA also had a lower serotype specificity of 52% compared to the other serotypes, where 10/21 of the heterologous sera generated positive signals (4/7 for A, 3/7 for SAT2, and 3/7 for O). On the other hand, the highest serotype specificity of 89% (1/5 for SAT1 and 1/7 for O) for VLP ELISA was seen for serotype A, while the other two VLP antigens had a lower serotype specificity, serotype O at 84% (3/7 of A sera reacted) and 79% for serotype SAT2 (1/7 for A reacted and 2/7 for O reacted). In pELISA, serotypes O, A, and SAT2 had a serotype specificity of 79%, 4/5 for SAT1, 3/5 for SAT1 and 1/7 for O, and 1/5 for SAT1 and 3/7 for O, respectively. These data for each of the ELISAs are summarized in Table S1, together with the data for VNT testing of the same sera.

## Correlation between VNT and ELISA signals generated with the peptides and VLPs

Results generated with the new ELISAs were compared to the corresponding titers measured using VNT (Fig. 5). These analyses revealed cases where samples with the highest and lowest VNT titers gave the strongest and weakest signals in the corresponding peptide or VLP ELISAs. The number of positive samples in VLPs correlated better with VNT than pELISA in the case of serotypes A and SAT2, whereas the number of positives was the same in the case of serotypes O and SAT1 when compared with VNT results. There was a trend in that the signals reported in the VLP ELISAs were higher and closer to each other than the signals reported by pELISA, which was weaker and dispersed. These relationships for the different serotypes were influenced by the number of samples and the extent of positivity determined in the VNT.

O KEN/4/2018                                          A SUD/9/2018

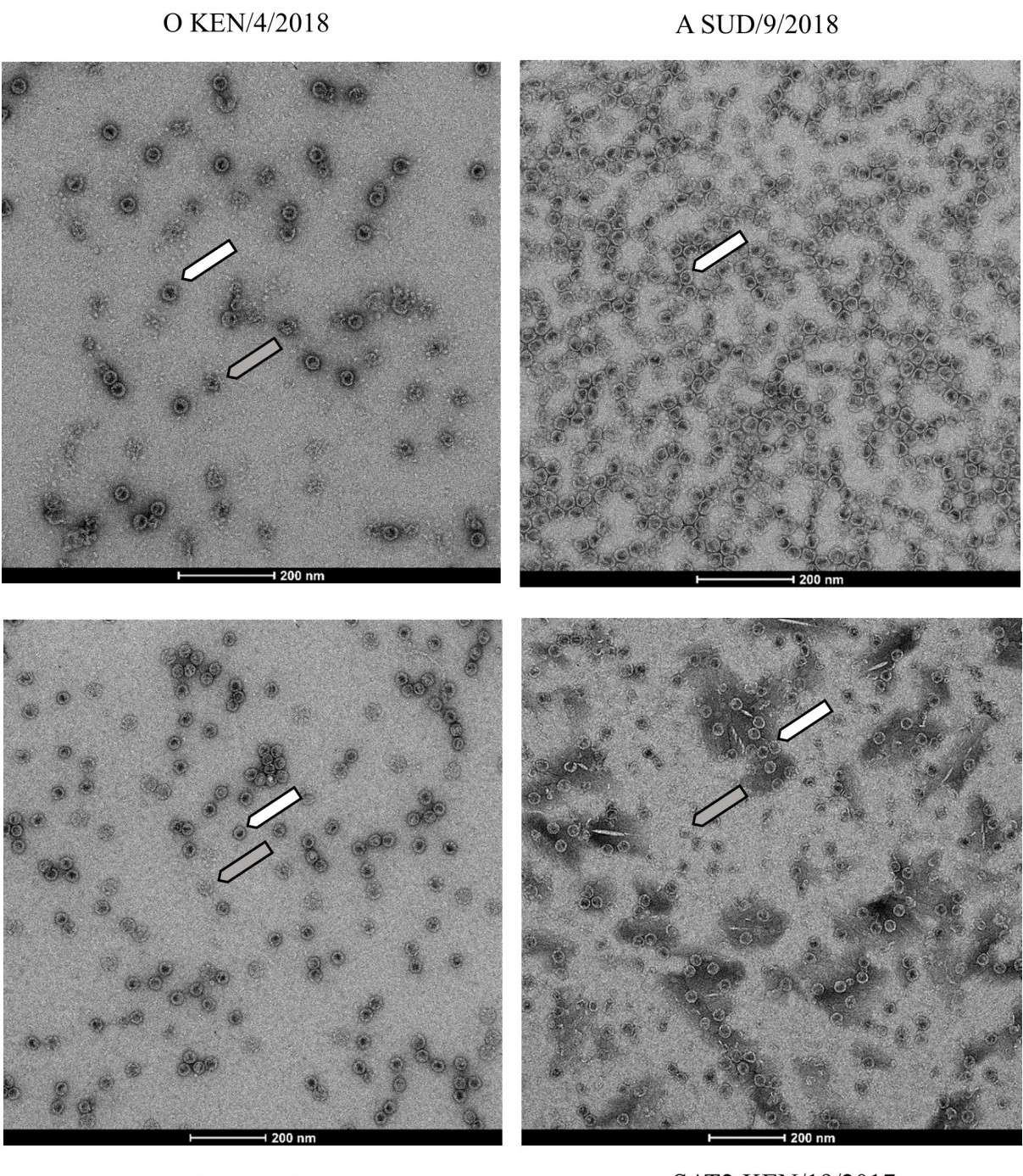

SAT1 TAN/22/2014                                     SAT2 KEN/19/2017

**FIG 2** Electron microscope images of the produced VLPs. O KEN/4/2018, A SUD/9/2018, SAT1 TAN/22/2014, and SAT2 KEN/19/2017 with examples of 75S (full capsid—white arrow) and 12S (disassociated pentamers—gray arrow) highlighted. Scale bars are shown for each electron micrograph (scale bar represents 200 nm).

## DISCUSSION

In FMD-endemic regions where often more than one serotype is present, surveillance and post-vaccine monitoring are crucial for FMD control. The currently available commercial kits and in-house serological ELISAs suffer from cross-reactivity, which could

**TABLE 2** Summary of the reactivity of the bovine sera against each FMDV isolate in pELISA, VLP ELISA, and VNT[a]

| Sera | O KEN/4/2018 | | | A SUD/9/2018 | | | SAT1 TAN/22/2014 | | | SAT2 KEN/19/2017 | | |
|---|---|---|---|---|---|---|---|---|---|---|---|---|
| | pELISA | VLP ELISA | VNT | pELISA | VLP ELISA | VNT | pELISA | VLP ELISA | VNT | pELISA | VLP ELISA | VNT |
| O | 6/7 | 6/7 | 7/7 | 1/7 | 1/7 | 0/7 | 5/7 | 3/7 | 0/7 | 3/7 | 2/7 | 2/7 |
| A | 0/7 | 3/7 | 0/7 | 5/7 | 6/7 | 4/7 | 1/7 | 4/7 | 0/7 | 0/7 | 1/7 | 1/7 |
| SAT1 | 4/5 | 0/5 | 2/5 | 3/5 | 1/5 | 0/5 | 5/5 | 5/5 | 4/5 | 1/5 | 1/5 | 0/5 |
| SAT2 | 0/7 | 0/7 | 0/7 | 0/7 | 0/7 | 0/7 | 0/7 | 3/7 | 0/7 | 6/7 | 7/7 | 6/7 |

[a]Gray shading denotes strain cross-reactivity.

be due to common external or internal epitopes being exposed when the full capsid is used as an antigen in these serological ELISAs (42).

The G-H loop is equivalent to the receptor-binding domain (RBD) for other viruses, where it has been used to develop specific immunoassays, particularly for viruses that exhibit antigenic diversity, such as SARS-CoV-2 (43, 44), dengue virus, and Zika virus. RBD is a critical surface-exposed region that is central to the neutralization responses of FMDV and is likely to play an important contribution to the serotype phenotype of FMDV isolates, making it a potentially ideal target for serotype-specific immunoassays (45). We hypothesized that the G-H loop contains serotype-specific epitope(s), providing a simple approach to present antigens for serological tests.

VNT is considered the gold standard for FMDV sero-diagnosis and vaccine matching as it measures the ability of the test serum to neutralize the virus. However, our data demonstrate that the serotype specificity of VNT is less than 100%, as inter-serotype cross-neutralization was observed with some of the sera. Similar findings have been reported previously for cross-reactive responses between serotypes O and A (18), and a cross-neutralizing epitope has been identified for O and A (46).

In this study, full-length G-H loop peptides were tested as antigens in ELISAs in comparison to full capsids in the form of VLPs. Although this pilot study only tested relatively small numbers of sera, the serotype sensitivity of the pELISAs and VLP ELISAs was comparable. For example, in both assay formats, 100% of the homologous sera

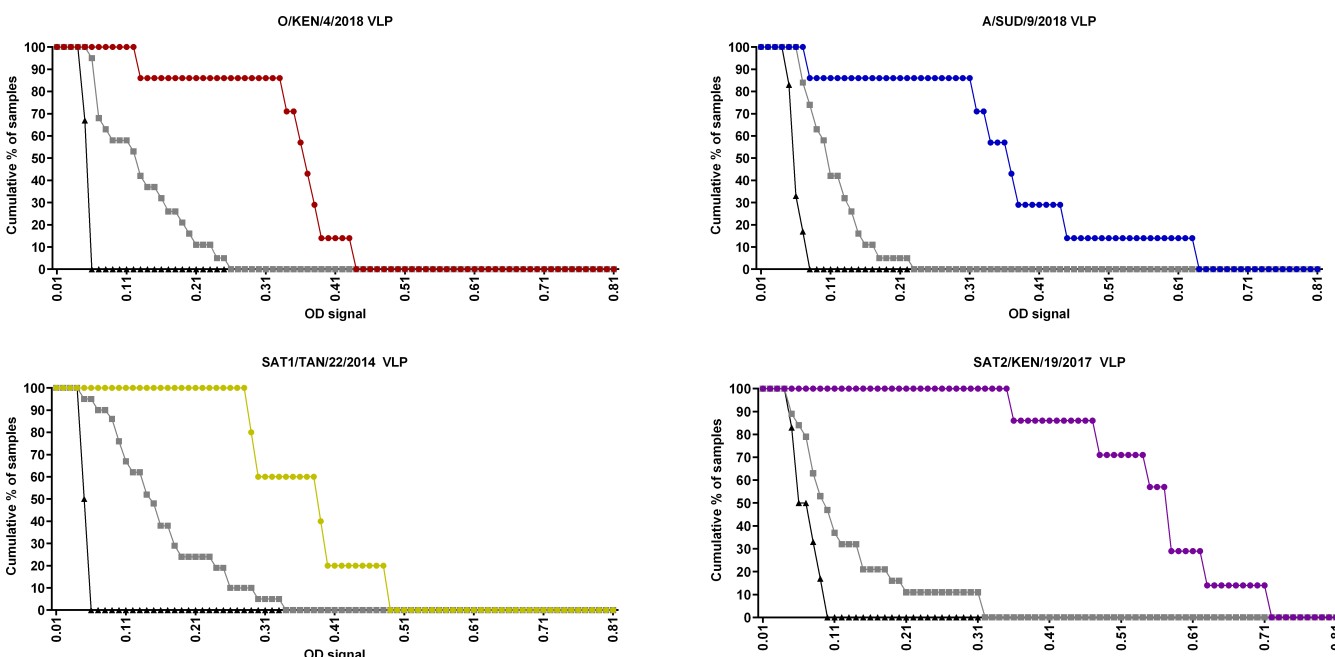

**FIG 3** Cumulative reactivity of the sera tested in the VLP ELISAs; O KEN/4/2018, A SUD/9/2018, SAT1 TAN/22/2014, and SAT2 KEN/19/2017 tested against a set of monovalent sera at a 1/125 dilution: O (red), A (blue), SAT1 (yellow), and SAT2 (purple). Responses of heterologous sera are highlighted as gray lines. while those for negative sera are in black (Table S1).

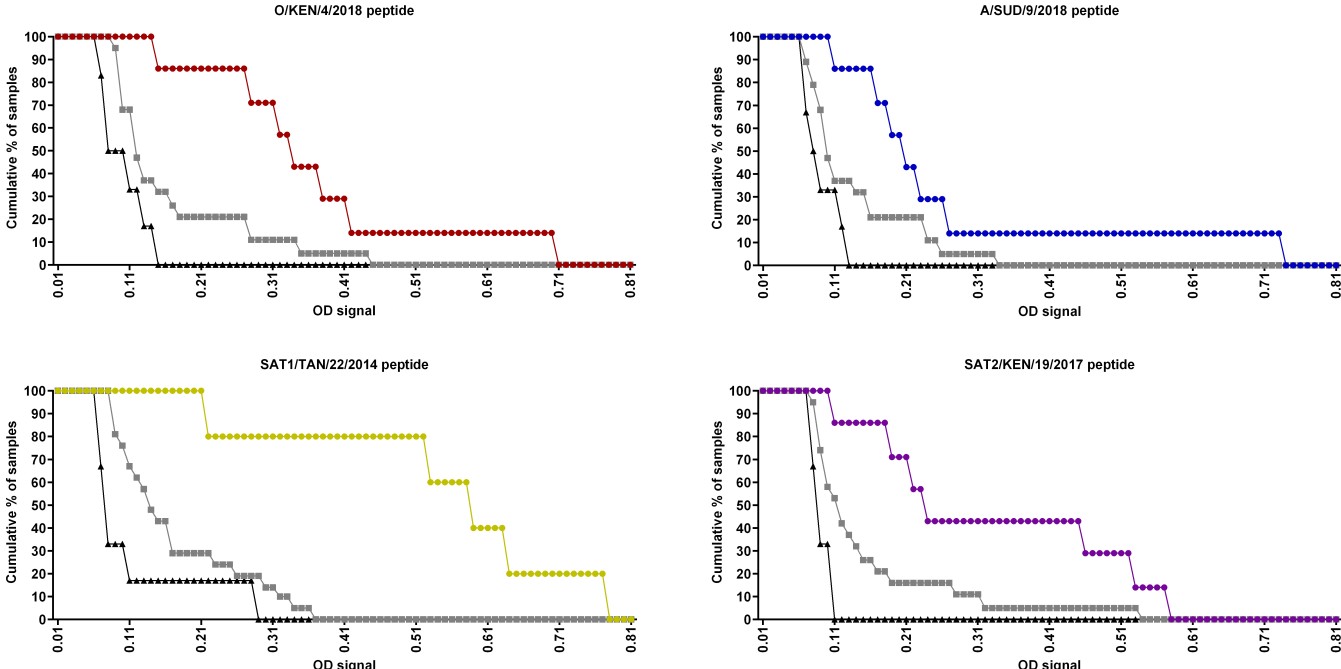

**FIG 4** Cumulative reactivity of the sera tested in the pELISAs; O KEN/4/2018, A SUD/9/2018, SAT1 TAN/22/2014, and SAT2 KEN/19/2017 tested in ELISA format against a set of monovalent sera at a 1/25 dilution: O (red), A (blue), SAT1 (yellow), and SAT2 (purple). Responses of heterologous sera are highlighted as gray lines, while those for negative sera are in black (Table S1).

reacted to the SAT1 TAN/22/2014 antigen and 86% (6/7) reacted to O KEN/4/2018. For SAT2 KEN/19/2017, a slightly lower serotype sensitivity was seen in the pELISA of 86%

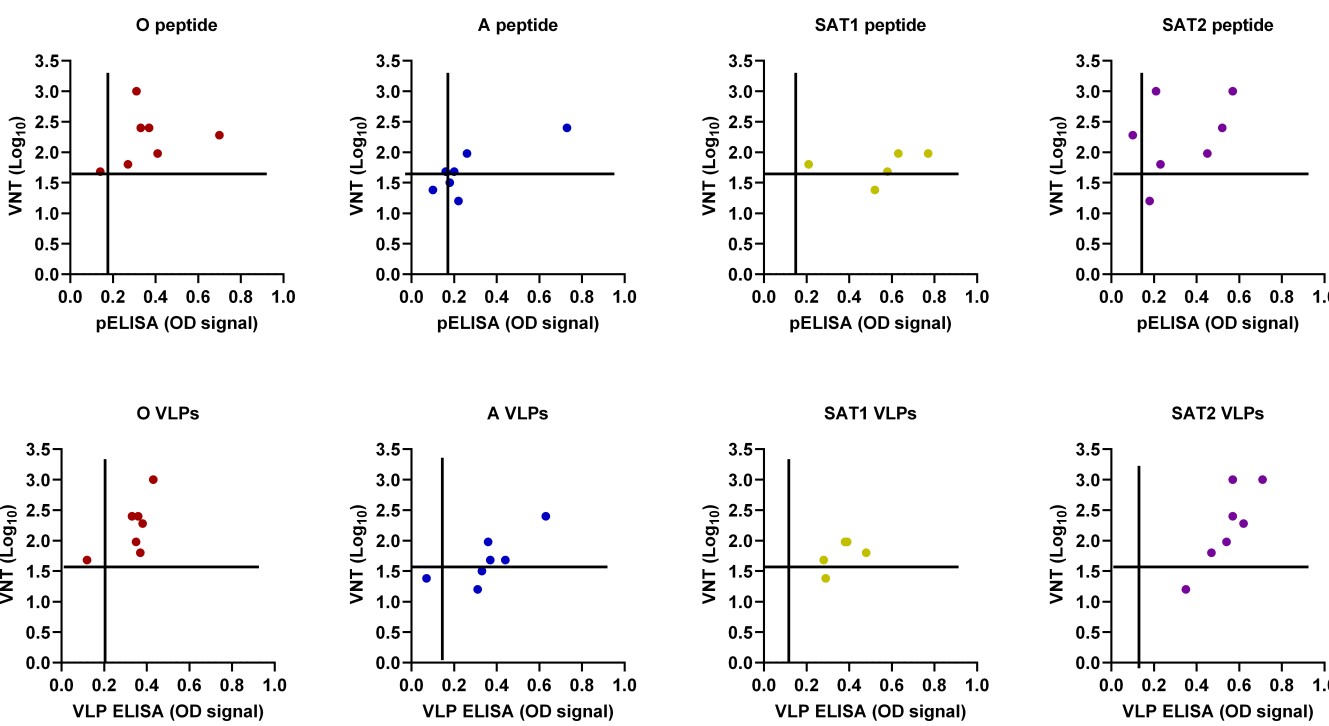

**FIG 5** Correlation between VNT and pELISA and VLP ELISA. Plots show data for serotypes O (red), A (blue), SAT1 (yellow), and SAT2 (purple). The horizontal and vertical black lines indicate the cutoffs established for VNT and VLP and peptide ELISAs.

(6/7) compared to 100% (7/7) for the VLP ELISA, while for A SUD/9/2018, 71% (5/7) and 86% (6/7) of serotype sensitivity were recorded for pELISA and VLP ELISA, respectively.

The serotype specificity of these tests was defined by the propotion of heterologous sera that reacted in the VLP and peptide ELISAs. The lowest observed serotype specificity of 71% for the pELISAs was for SAT1; the corresponding SAT1 VLP ELISA also had a lower serotype specificity of 52% compared to the other serotypes perhaps due to the presence of a high proportion of degraded 12S particles revealed by electron microscopy when compared to the other VLPs. In contrast, the highest serotype specificity of 89% (1/5 for SAT1 and 1/7 for O) for VLP ELISA was seen for serotype A, which appeared to have the highest proportion of intact 75S particles (Fig. 2). The other two VLP antigens have a lower serotype specificity, serotype O has serotype specificity at 84% and SAT2 has serotype specificity at 79%. In pELISA, serotypes O, A, and SAT2 had a serotype specificity of 79%, respectively.

All peptide antigens for serotypes O, A, and SAT1 shared a RGDL motif, while the SAT2 peptide had a RGDR motif. This "RGDL" motif is considered a main target of porcine cross-serotype broadly neutralizing antibodies (bnAbs) (47). The presence of this common motif could explain the cross-neutralization seen by some samples against O KEN/4/2018. Nevertheless, no cross-neutralization was seen between serotype A and SAT1 responses despite the presence of RGDL in both peptides. Previously, a cross-reactive epitope has been identified associated with the RGD motif (XRGDX) in serotypes O, A, and SAT1 (48). However, the sequence alignments did not provide a simple explanation to the cross-reactivity seen or even explain the serotype sensitivity against the homologous sera; however, these responses could be due to the different folding and presentation of the G-H loop epitope/s resulting in different orientations when they are used to coat the ELISA plate, hence, different patterns of recognition by antibodies in the sera. Even with this cross-reactivity seen, no neutralization was seen against A and SAT1, which highlights that binding in ELISA and neutralization are distinct properties of the sera.

The peptides, VLPs, and VNT viruses were not necessarily matched to the specific antigens that the cattle had been exposed to. Hamblin et al. mentioned that ELISA is more sensitive and reproducible than the VNT (2, 9). This statement is in line with our results; the serotype sensitivity of A SUD/9/2018, SAT1 TAN/22/2014, and SAT2 KEN/19/2017 in VNT was lower than that of the corresponding pELISAs and VLP ELISAs. Further investigation is required by testing more sera, especially against SAT1 sera as all SAT1 serum samples were obtained after vaccination with a single SAT1 isolate. Five of the sera failed to generate titers in the homologous VNT systems, three for A and single sera for SAT1 and SAT2. Among these sera, two serotype A sera and the single sera for SAT1 and SAT2 reacted in both the corresponding pELISA and VLP ELISA, suggesting that these assays may be able to detect non-neutralizing antibodies, increasing their serotype sensitivity. Since the pELISAs were expected to present a focused repertoire of epitopes surrounding the RGD motif, which is an important site for neutralization activity, it might be expected that the results for these ELISAs would more closely conform to the VNT data compared to the VLP ELISAs where a wide range of neutralizing and non-neutralizing sites were anticipated to be presented. However, the data generated from these comparisons between the tests did not support this hypothesis.

In conclusion, the G-H loop peptides could be potential candidates for novel serological assays. These peptides provide a biosafe and cost-effective approach to present and engineer FMDV-specific antigens that compare favorably to using full VLP capsid antigens in ELISAs. However, further validation with a larger number of strains and sera would need to be undertaken, where the development of alternative ELISA formats such as liquid-phase blocking tests might improve the test performance.

## ACKNOWLEDGMENTS

The PhD studies of AAY have been financially supported by Newton Mosharafa Fund (Bureau ID: NMM11/21) and the Egyptian Ministry of Higher Education and Scientific

Research, Cultural Affairs and Mission Sector, Egypt, and The Pirbright Institute for performing all the practical work. Work at Pirbright is funded by the UK Department for Environment, Food and Rural Affairs (projects SE1131 and SE1132). The Pirbright Institute also receives grant‐aided support from the Biotechnology and Biological Sciences Research Council (BBSRC) of the United Kingdom (projects BBS/E/I/00007037, BB/X011038/1, BB/X011046/1, BBS/E/PI/230002C, and BBS/E/PI/23NB0004).

## AUTHOR AFFILIATIONS

[1]The Pirbright Institute, Pirbright, United Kingdom

[2]School of Veterinary Medicine, Department of Comparative Biomedical Sciences, University of Surrey, Guildford, United Kingdom

[3]Department of Foot and Mouth Disease, Veterinary Serum and Vaccine Research Institute, Cairo, Egypt

[4]Animal and Plant Health Agency, Virology Department, Weybridge, United Kingdom

## PRESENT ADDRESS

Madeeha Afzal, Sir William Dunn School of Pathology, University of Oxford, Oxford, United Kingdom

## AUTHOR ORCIDs

Abdelaziz A. Yassin  http://orcid.org/0000-0003-4657-2277
Madeeha Afzal  http://orcid.org/0000-0001-8954-8934
Donald P. King  http://orcid.org/0000-0002-6959-2708
Amin S. Asfor  http://orcid.org/0009-0000-3493-2576

## FUNDING

| Funder | Grant(s) | Author(s) |
| --- | --- | --- |
| Biotechnology and Biological Sciences Research Council | BB/X011038/1 | Donald P. King |
| Biotechnology and Biological Sciences Research Council | BB/X011046/1 | Donald P. King |
| Biotechnology and Biological Sciences Research Council | BBS/E/I/00007037 | Donald P. King |
| Biotechnology and Biological Sciences Research Council | BBS/E/PI/230002C | Donald P. King |
| Biotechnology and Biological Sciences Research Council | BBS/E/PI/23NB0004 | Donald P. King |
| Department for Environment, Food and Rural Affairs, UK Government | SE1131 | Donald P. King |

## AUTHOR CONTRIBUTIONS

Abdelaziz A. Yassin, Data curation, Formal analysis, Methodology, Validation, Writing – original draft | Anna B. Ludi, Methodology, Project administration, Supervision, Validation, Writing – review and editing | Alison Burman, Methodology, Resources, Writing – review and editing | Georgina Limon, Methodology, Writing – review and editing | Madeeha Afzal, Methodology, Writing – review and editing | Daniel Horton, Investigation, Project administration, Supervision, Writing – review and editing | Donald P. King, Conceptualization, Funding acquisition, Resources, Supervision, Visualization, Writing – review and editing | Amin S. Asfor, Conceptualization, Funding acquisition, Investigation, Methodology, Resources, Supervision, Visualization, Writing – review and editing.

## ADDITIONAL FILES

The following material is available online.

### Supplemental Material

**Figure S1 and Table S1 (Spectrum03514-25-s0001.docx).** ROC analysis of sera reactivity and a list of tested bovine sera,

### Open Peer Review

**PEER REVIEW HISTORY (review-history.pdf).** An accounting of the reviewer comments and feedback.

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
