## [Reviewer comments · Microbiology Spectrum]

Microbiology Spectrum

Performance of prototype serological immunoassays for foot-and-mouth disease virus using G-H loop peptides and stabilised virus-like particles

Abdelaziz A. Yassin, Yvonne Sewell, Anna Ludi, Alison Burman, Georgina Limon, Madeeha Afzal, Daniel Horton, Donald King, and Amin Asfor

Corresponding Author(s): Amin Asfor, University of Surrey

Review Timeline:

Submission Date:	November 4, 2025
Editorial Decision:	December 12, 2025
Revision Received:	February 27, 2026
Accepted:	March 9, 2026

Editor: Alexander Bello

Reviewer(s): The reviewers have opted to remain anonymous.

Transaction Report:

DOI: <https://doi.org/10.1128/spectrum.03514-25>

Re: Spectrum03514-25 (Performance of prototype serological immunoassays for foot-and-mouth disease virus using G-H loop peptides and stabilised virus-like particles)

Dear Dr. Amin Asfor:

Thank you for the privilege of reviewing your work. Below you will find my comments, instructions from the Spectrum editorial office, and the reviewer comments.

Revision Guidelines

Sincerely,
Alexander Bello
Editor
Microbiology Spectrum

Reviewer #1 (Comments for the Author):

Manuscript ID: Spectrum03514-25

"Performance of prototype serological immunoassays for foot-and-mouth disease virus using G-H loop peptides and stabilised virus-like particles"

This manuscript evaluated synthetic peptides encompassing the hypervariable G-H loop of foot-and-mouth disease viruses (FMDVs) currently circulating in East Africa (O, A, SAT1 and SAT2), as capture antigens in ELISAs (pELISAs) intended as serotype-specific tests. For this, a panel of monovalent bovine sera from known origin were tested in parallel with pELISAs, ELISAs using stabilized FMDV VLPs, and Virus neutralisation tests (VNT) as benchmark test.

The results obtained indicated the sensitivity of the evaluated tests ranged from 71-100% and 86-100% for the pELISA and VLP ELISA formats, respectively; while the specificity ranged from 71-79% and 52-89% for the pELISA and VLP ELISAs, respectively. The corresponding results for VNT assays were: 57-100% sensitivity and 84-100% specificity. From these results the authors conclude that G-H loop peptides offer a simple, bio-safe and cost-effective approach to present FMDV-specific epitopes, and that these initial findings suggest that pELISAs could represent a promising approach for the development of serotype-specific FMDV ELISA assays

Main comments:

- I am not sure that the proposed pELISA can really serve as a serotype-specific serological test, given the results shown. I understand it is not an easy task, since the benchmark test (VNT) does not offer high specificity either (84-100% specificity). On the other hand, I appreciate that if the proposed pELISA can offer results similar to VNT, it would represent an interesting improvement, as it is a simpler and faster test, that doesn't require handling infective FMDV in high containment laboratories.

- Could it be possible to improve the results obtained by the pELISA using peptides conjugated to biotin in an ELISA biotin-streptavidin assay? Or in liquid phase blocking tests?

Other comments:

- Lines 259-260: "The lowest observed specificity of 71% for the pELISAs was for SAT 1 where 7/21 heterologous sera cross-reacted to the peptide". This calculation seems to be incorrect. If 7/21 of the heterologous sera cross-reacted to the peptide, the corresponding specificity would be 66%. Thus, the overall results for the pELISAs would be: sensitivity ranging 71-100% and specificity ranging 66-79%

- Table 1. Peptides corresponding to serotypes O and A were 25 amino acids long, SAT1 peptide was 26 amino acids long and SAT 2 peptide was 29 amino acids long. What were the reasons for these differences? Were they due to structural considerations regarding differences in the conformation of the G-H loop? I think it would be interesting to explain this.

- Line 130: "Thirty-one bovine sera, seven for serotype A, O and SAT2 and five for SAT1". This is incorrect. The total sum of the indicated sera is 26, to which 6 negative sera were added as negative control samples, for a total of 32. Please, correct.

- Figure 2. I think it would be helpful to label the panels with the names of the viral strains instead of a, b, c, d, so the reader doesn't have to go through the legend to identify the different VLPs.

- Figures 3 and 4 are not described in text, while the results reported in section 4.3 (lines 194-210) are based in the data collected in Table S1, shown as supplementary material. This should be corrected. I think Table S1 should be incorporated in the manuscript and figures 3 and 4 should be commented in the text or else moved to supplementary materials.

- Table S1. I think it would help readers to keep the same order in the display of the different serotypes as in the rest of the figures and tables shown: i.e., serotypes O, A, SAT 1 and SAT 2

- Lines 222-223: "Results generated with the new ELISAs were compared to the corresponding titres measured using VNT". At the end of this sentence indicate this results are shown in Figure 5 (which is not mentioned in the manuscript text).

Reviewer #2 (Comments for the Author):

Please consider these suggestions.

Lines 83-85:

'The sensitivity (the number of homologous sera that were correctly detected using the peptides or VLPs) and the specificity (the number of heterologous sera that falsely reacted against the peptide or VLPs)'

Also, lines 161-164:

'Sensitivity was calculated as the number of homologous sera (one serotype for all the settings) that reacted against the peptides, VLPs or VNT. Specificity was calculated as the number of heterologous (serotype other than the tested serotype) sera that reacted against the peptides, VLPs or in the VNT.'

- The sensitivity of the test is the proportion of sampled positives that tested positive, while the specificity is the proportion of sampled negatives that tested negative. The parameter in this study should be called 'serotype specificity'. It should be the rate or proportion, not the number of sera. The wording in the Methods implies the 'specificity' was higher if there were more heterologous sera reacting against the mismatched peptides or VLPs. The description likely yields the measure of cross-reactivity, not of serotype specificity.

Lines 257-267:

'The lowest observed specificity of 71% for the pELISAs was for SAT 1 where 7/21 heterologous sera cross-reacted to the peptide (1/7 for A, 1/7 for SAT2 and 5/7 for O). <...>'

- The values of 'specificity' are inconsistent with the method described in Lines 161-164. Please use the precise terminology and rewrite the definitions Introduction and Methods, and the calculations in Methods.

Lines 33-34:

'The Specificity of these tests ranged from 71-79% and 52-89% for the pELISA and VLP ELISA formats, respectively.'

- The 52% 'specificity' is not derived in the text, only present in the Abstract. The terminology may be misleading, please use 'serotype specificity' in case the Authors meant it.

Line 97:

'at a TCID₅₀ virus dose between 32-320 (1.5log₁₀ to 2.5log₁₀)'

- the notation may represent common usage, however, it lacks mathematical integrity. It would benefit from adding the units and treating the log₁₀ as in 'virus dose between 32-320 particles/mL (1.5 to 2.5 log₀ particles/mL)' (please use the applicable).

Lines 164-165:

'The overall test accuracy was calculated by summing the sensitivity and the specificity data divided by 2'

- The accuracy was not mentioned in the Results.

Line 191:

'Figure 2: Electron microscope images of the produced VLPs at the magnification of 200 nm'

- the bars under the images represent 200 nm under the given magnification, which is not listed in the caption.

Lines 202-203:

'Only four serotype SAT1 sera cross-reacted with serotype O pELISA'

- 'only' is excessive here.

Lines 223-226:

'These analyses revealed a trend where samples with the highest and lowest VN titres gave strongest and weakest signals in most of the corresponding peptide or VLP ELISAs. These relationships for the different serotypes were influenced by the numbers of samples and extent of positivity determined in the VNT.'

- Is the graphical presentation in Fig. 5 the only analysis supporting the conclusions? Trend lines can be implied, however, they are not depicted. One can speculate the SAT1 peptide and VLP graphs represent the weakest correlation, however the relation among the graphs and the statements is not well described.

Lines 287-289:

'Five of the sera failed to generate titres in the homologous VNT systems; three for A and single sera for SAT1 and SAT2. Two serotype A sera and the single sera for SAT1 and SAT2 reacted in both the corresponding pELISA and VLP ELISA <...>'

- please clarify, e. g.: 'Among these sera, two serotype A sera and the single sera for SAT1 and SAT2 <././>'

Please consider these suggestions.

Lines 83–85:

‘The sensitivity (the number of homologous sera that were correctly detected using the peptides or VLPs) and the specificity (the number of heterologous sera that falsely reacted against the peptide or VLPs)’

Also, lines 161–164:

‘Sensitivity was calculated as the number of homologous sera (one serotype for all the settings) that reacted against the peptides, VLPs or VNT. Specificity was calculated as the number of heterologous (serotype other than the tested serotype) sera that reacted against the peptides, VLPs or in the VNT.’

– The sensitivity of the test is the proportion of sampled positives that tested positive, while the specificity is the proportion of sampled negatives that tested negative. The parameter in this study should be called ‘serotype specificity’. It should be the rate or proportion, not the number of sera. The wording in the Methods implies the ‘specificity’ was higher if there were more heterologous sera reacting against the mismatched peptides or VLPs. The description likely yields the measure of cross-reactivity, not of serotype specificity.

Lines 257–267:

‘The lowest observed specificity of 71% for the pELISAs was for SAT 1 where 7/21 heterologous sera cross-reacted to the peptide (1/7 for A, 1/7 for SAT2 and 5/7 for O). <...>’

– The values of ‘specificity’ are inconsistent with the method described in Lines 161–164. Please use the precise terminology and rewrite the definitions Introduction and Methods, and the calculations in Methods.

Lines 33–34:

‘The Specificity of these tests ranged from 71-79% and 52-89% for the pELISA and VLP ELISA formats, respectively.’

– The 52% ‘specificity’ is not derived in the text, only present in the Abstract. The terminology may be misleading, please use ‘serotype specificity’ in case the Authors meant it.

Line 97:

‘at a TCID₅₀ virus dose between 32-320 (1.5log₁₀ to 2.5log₁₀)’

– the notation may represent common usage, however, it lacks mathematical integrity. It would benefit from adding the units and treating the log₁₀ as in ‘virus dose between 32–320 particles/mL (1.5 to 2.5 log₁₀ particles/mL)’ (please use the applicable).

Lines 164–165:

‘The overall test accuracy was calculated by summing the sensitivity and the specificity data divided by 2’

– The accuracy was not mentioned in the Results.

Line 191:

‘Figure 2: Electron microscope images of the produced VLPs at the magnification of 200 nm’

– the bars under the images represent 200 nm under the given magnification, which is not listed in the caption.

Lines 202–203:

‘Only four serotype SAT1 sera cross-reacted with serotype O pELISA’

– ‘only’ is excessive here.

Lines 223–226:

‘These analyses revealed a trend where samples with the highest and lowest VN titres gave strongest and weakest signals in most of the corresponding peptide or VLP ELISAs. These relationships for the different serotypes were influenced by the numbers of samples and extent of positivity determined in the VNT.’

– Is the graphical presentation in Fig. 5 the only analysis supporting the conclusions? Trend lines can be implied, however, they are not depicted. One can speculate the SAT1 peptide and VLP graphs represent the weakest correlation, however the relation among the graphs and the statements is not well described.

Lines 287-289:

‘Five of the sera failed to generate titres in the homologous VNT systems; three for A and single sera for SAT1 and SAT2. Two serotype A sera and the single sera for SAT1 and SAT2 reacted in both the corresponding pELISA and VLP ELISA <...>’

– please clarify, e. g.: ‘Among these sera, two serotype A sera and the single sera for SAT1 and SAT2 <../.>’

Response to the Reviewers` Comments

Reviewer #1 (Comments for the Author):

Manuscript ID: Spectrum03514-25

Main comments:

- I am not sure that the proposed pELISA can really serve as a serotype-specific serological test, given the results shown. I understand it is not an easy task, since the benchmark test (VNT) does not offer high specificity either (84-100% specificity). On the other hand, I appreciate that if the proposed pELISA can offer results similar to VNT, it would represent an interesting improvement, as it is a simpler and faster test, that doesn't require handling infective FMDV in high containment laboratories.

Q:- Could it be possible to improve the results obtained by the pELISA using peptides conjugated to biotin in an ELISA biotin-streptavidin assay? Or in liquid phase blocking tests?

A: Thank you for the suggestion. We evaluated biotinylated peptides as a preliminary test to determine whether they offered any improvement over the non-biotinylated form. However, no differences were observed between the two approaches, indicating that biotinylation did not enhance assay sensitivity or specificity. Hence, we proceeded with the non-biotinylated peptides to minimise cost without compromising performance. The liquid-phase blocking ELISA was not included in our evaluation. We therefore recommend that future studies assess LPBE to determine whether it can further improve test performance. (Please see lines 311 to 312 on page 12)

Other comments:

Q- Lines 259-260: "The lowest observed specificity of 71% for the pELISAs was for SAT 1, where 7/21 heterologous sera cross-reacted to the peptide". This calculation seems to be incorrect. If 7/21 of the heterologous sera cross-reacted to the peptide, the corresponding specificity would be 66%. Thus, the overall results for the pELISAs would be sensitivity ranging 71-100% and specificity ranging 66-79%

A: Thank you for the comment. The number of heterologous sera that cross-reacted with the peptide was 6/21, not 7/21. The specificity percentage for pELISA for SAT1 has been corrected accordingly. Please see page (8), line (217, 218):

The new corrected text 'The lowest observed serotype specificity of 71% for the pELISAs was for SAT1, where 6/21 heterologous sera cross-reacted to the peptide

(1/7 for A, and 5/7 for O).”

Q: Table 1. Peptides corresponding to serotypes O and A were 25 amino acids long, SAT1 peptide was 26 amino acids long and SAT 2 peptide was 29 amino acids long. What were the reasons for these differences? Were they due to structural considerations regarding differences in the conformation of the G-H loop? I think it would be interesting to explain this.

A: Thanks for your suggestion.

- This paragraph has been added in the introduction page number (4), line number (79-85)

The length of VP1 is variable between serotypes due to insertions or deletions mainly in the region around the G-H loop [31]. The G–H loop is a highly flexible, surface-exposed region of the capsid, and its conformation differs slightly between serotypes, influencing the range of loop lengths that can be structurally accommodated [32, 33]. Comparative genomic analyses show that SAT serotypes, in particular, exhibit length variability in this region, largely due to their distinct evolutionary histories and circulation in wildlife reservoirs (Bastos et al., 2001; Carrillo et al., 2005).

- The following sentence was added in the M&M on page (5), line number (115-116), to justify the variable length of the designed peptides

“The differences in the peptide lengths design reflect the difference in the conformation of the G-H loop between different serotypes”.

Q: Line 130: "Thirty-one bovine sera, seven for serotype A, O and SAT2 and five for SAT1". This is incorrect. The total sum of the indicated sera is 26, to which 6 negative sera were added as negative control samples, for a total of 32. Please, correct.

A: Many thanks for the comment. This has been corrected. Please see line 142, Page no.6

“**Thirty-two bovine** sera, seven for serotype A, O and SAT2 and five for SAT1 were selected from animals infected and/or vaccinated with a single FMDV serotype (i.e. monovalent) representing O, A, SAT 1, or SAT 2 (Table S1)”

Q:- Figure 2. I think it would be helpful to label the panels with the names of the viral strains instead of a, b, c, d, so the reader doesn't have to go through the legend to identify the different VLPs.

A: Thanks for the suggestion. The names of the virus strains have been added.

Q- Figures 3 and 4 are not described in text, while the results reported in section 4.3 (lines 194-210) are based in the data collected in Table S1, shown as supplementary material. This should be corrected. I think Table S1 should be incorporated in the manuscript and figures 3 and 4 should be commented in the text or else moved to

supplementary materials.

A: Thanks for the suggestion. It is very helpful. The table has been incorporated in the manuscript, and figures 3 and 4 are now commented in the text line (228), page (9).

Q- Table S1. I think it would help readers to keep the same order in the display of the different serotypes as in the rest of the figures and tables shown: i.e., serotypes O, A, SAT 1, and SAT 2

A: Thanks for your suggestion. The order has been modified. It is now labelled as Table (2) on page (9)

Q: Lines 222-223: "Results generated with the new ELISAs were compared to the corresponding titres measured using VNT". At the end of this sentence, indicate this results are shown in Figure 5 (which is not mentioned in the manuscript text.

A: Thanks. It has been added. Please see line 233-239 in page 9

Reviewer #2 (Comments for the Author):

Please consider these suggestions.

Lines 83-85:

Q: 'The sensitivity (the number of homologous sera that were correctly detected using the peptides or VLPs) and the specificity (the number of heterologous sera that falsely reacted against the peptide or VLPs)'

Also, lines 161-164:

'Sensitivity was calculated as the number of homologous sera (one serotype for all the settings) that reacted against the peptides, VLPs or VNT. Specificity was calculated as the number of heterologous (serotype other than the tested serotype) sera that reacted against the peptides, VLPs or in the VNT.'

- The sensitivity of the test is the proportion of sampled positives that tested positive, while the specificity is the proportion of sampled negatives that tested negative. The parameter in this study should be called 'serotype specificity'. It should be the rate or proportion, not the number of sera. The wording in the Methods implies the 'specificity' was higher if there were more heterologous sera reacting against the mismatched peptides or VLPs. The description likely yields the measure of cross-reactivity, not of serotype specificity.

A: Many thanks for the comment: the term serotype specificity was introduced to clarify the definition of specificity in this context.

Lines 257-267:

'The lowest observed specificity of 71% for the pELISAs was for SAT 1 where 7/21 heterologous sera cross-reacted to the peptide (1/7 for A, 1/7 for SAT2 and 5/7 for O). <...>'

Q: The values of 'specificity' are inconsistent with the method described in Lines 161-164. Please use the precise terminology and rewrite the definitions in the Introduction and Methods, and the calculations in Methods.

A: Many thanks for your comment, the specificity definition was revised and replaced by the term serotype specificity to reflect the ability of the test to correctly recognise the sera raised against the same serotype verse the cross reactivities. The serotype specificity (the proportion of heterologous sera that did not react against the peptide or VLPs)

The revised definition was inserted on page number 4 and line number 94-97

Lines 33-34:

'The Specificity of these tests ranged from 71-79% and 52-89% for the pELISA and VLP ELISA formats, respectively.'

Q: - The 52% 'specificity' is not derived in the text, only present in the Abstract. The terminology may be misleading, please use 'serotype specificity' in case the Authors meant it.

A: Thanks for your comment. Specificity was replaced by serotype specificity wherever applicable.

-The 52% 'specificity' is now in the results section as above and also mentioned in the discussion section page (8), line no. (219)

“The corresponding SAT 1 VLP ELISA also had a lower serotype specificity of 52% compared to the other serotypes, where 10/21 of the heterologous sera generated positive signals (4/7 for A, 3/7 for SAT2 and 3/7 for O)”

Q: Line 97:

'at a TCID₅₀ virus dose between 32-320 (1.5log₁₀ to 2.5log₁₀)'

- the notation may represent common usage, however, it lacks mathematical integrity. It would benefit from adding the units and treating the log₁₀ as in 'virus dose between 32-320 particles/mL (1.5 to 2.5 log₁₀ particles/mL)' (please use the applicable).

A: Many thanks for the observation. This has been modified to maintain mathematical integrity. Please see line 108 in page 5

“The results were reported as the final dilution required to neutralise 50% of the inoculated cultures [3] between 32-320 (1.5log₁₀ to 2.5log₁₀) TCID₅₀ virus doses. As stated by the World Organisation for Animal Health (WOAH), antibody titres below 1/45 (i.e. 1.65 log₁₀) of the final serum dilution were regarded as negative.”

Q: Lines 164-165:

'The overall test accuracy was calculated by summing the sensitivity and the specificity data divided by 2'

- The accuracy was not mentioned in the Results.

A: Thank you for the comment. The overall test accuracy had been calculated as the mean of sensitivity and specificity. We chose not to include this metric in the manuscript because sensitivity and serotype-specificity provide a more accurate and informative representation of assay performance. In addition, reporting a single '% accuracy' value could be misleading in this context, so this line was removed from the Methods section.

Q: Line 191:

'Figure 2: Electron microscope images of the produced VLPs at a magnification of 200 nm'

- The bars under the images represent 200 nm under the given magnification, which is not listed in the caption.

A: Thanks for the comment. The caption has been updated as follows:

“Figure 2: Electron microscope images of the produced VLPs. O KEN/4/2018, A SUD/9/2018, SAT1 TAN/22/2014 and SAT2 KEN/19/2017 with examples of 75S (full capsid - white arrow) and 12S (disassociated pentamers - grey arrow) highlighted. Scale bars are shown for each electron micrograph (Scale bar represents 200 nm).

Q: Lines 202-203:

'Only four serotype SAT1 sera cross-reacted with serotype O pELISA'

- 'only' is excessive here.

A: Many thanks for the comment. This has now been modified.

Please see line 208, page 8

“The sensitivity for the serotype O pELISA and VLP ELISA were both 86%. Four serotype SAT1 sera cross-reacted with serotype O pELISA and three serotype A sera cross-reacted with VLP ELISA”.

Q: Lines 223-226:

'These analyses revealed a trend where samples with the highest and lowest VN titres gave strongest and weakest signals in most of the corresponding peptide or VLP ELISAs. These relationships for the different serotypes were influenced by the numbers of samples and extent of positivity determined in the VNT.'

- Is the graphical presentation in Fig. 5 the only analysis supporting the conclusions? Trend lines can be implied, however, they are not depicted. One can speculate the SAT1 peptide and VLP graphs represent the weakest correlation, however the relation among the graphs and the statements is not well described.

A: Many thanks for the observation. Despite the number of samples used being small. More explanation was added to describe the case (These analyses revealed in some cases samples with the highest and lowest VN titres gave the strongest and weakest signals in most of the corresponding peptide or VLP ELISAs. The number of positive samples in VLPs is correlated better with VNT than pELISA in the case of Serotype A and SAT2, whereas the number of positive are the same in the case of serotype O and SAT1 when compared with VNT results. There is a trend in that the signals reported in the VLP ELISAs are higher and closer to each other than the signals reported by pELISA, which is weaker and dispersed.

Lines 287-289:

'Five of the sera failed to generate titres in the homologous VNT systems; three for A and single sera for SAT1 and SAT2. Two serotype A sera and the single sera for SAT1 and SAT2 reacted in both the corresponding pELISA and VLP ELISA <...>'

- please clarify, e. g.: 'Among these sera, two serotype A sera and the single sera for SAT1 and SAT2 <../.>'

Thanks for your suggestion. This has been modified as follows: lines 299 page 11

“Five of the sera failed to generate titres in the homologous VNT systems; three for A

and single sera for SAT1 and SAT2. Among these sera two serotype A sera and the single sera for SAT1 and SAT2 reacted in both the corresponding pELISA and VLP ELISA, suggesting that these assays may be able to detect non-neutralising antibodies, increasing their sensitivity.

Re: Spectrum03514-25R1 (Performance of prototype serological immunoassays for foot-and-mouth disease virus using G-H loop peptides and stabilised virus-like particles)

Dear Dr. Amin Asfor:

Your manuscript has been accepted, and I am forwarding it to the ASM production staff for publication. Your paper will first be checked to make sure all elements meet the technical requirements. ASM staff will contact you if anything needs to be revised before copyediting and production can begin. Otherwise, you will be notified when your proofs are ready to be viewed.

Sincerely,
Alexander Bello
Editor
Microbiology Spectrum

Reviewer #1 (Comments for the Author):

Manuscript ID: Spectrum03514-25R1

"Performance of prototype serological immunoassays for foot-and-mouth disease virus using G-H loop peptides and stabilised virus-like particles"

This is a revised version of the original manuscript. The authors have adequately addressed my suggestions. I have no further comments.

Reviewer #2 (Comments for the Author):

Thank you for taking into account the suggestions of the reviewers and improving the manuscript.